# Public Health Surveillance for Adverse Events Following COVID-19 Vaccination in Africa

**DOI:** 10.3390/vaccines10040546

**Published:** 2022-04-01

**Authors:** AbdulAzeez Adeyemi Anjorin, Ismail A. Odetokun, Jean Baptiste Nyandwi, Hager Elnadi, Kwame Sherrif Awiagah, Joseph Eyedo, Ajibola Ibraheem Abioye, George Gachara, Aala MohmedOsman Maisara, Youssef Razouqi, Mohamed Farah Yusuf Mohamud, Zuhal Ebrahim Mhgoob, Tunde Ajayi, Lazare Ntirenganya, Morounke Saibu, Babatunde Lawal Salako, Nusirat Elelu, Kikelomo Ololade Wright, Folorunso O. Fasina, Rasha Mosbah

**Affiliations:** 1Department of Microbiology (Virology Research), Lagos State University, Ojo 102101, Lagos, Nigeria; eyedoleejoseph@gmail.com; 2Department of Veterinary Public Health & Preventive Medicine, University of Ilorin, Ilorin 200213, Kwara State, Nigeria; odetokun.ia@unilorin.edu.ng (I.A.O.); elelu.n@unilorin.edu.ng (N.E.); 3Department of Pharmacy, University of Rwanda, Kigali 4285, Rwanda; nbaptiste1988@gmail.com; 4Infectious Diseases and One Health Department, Universite de Tours, 37000 Tours, France; hager.elnadi@gmail.com; 5Training and Research Unit, Korle Bu Teaching Hospital, Accra P.O. Box KB 77, Ghana; awiagahsherrif@gmail.com; 6Department of Nutrition, Harvard University, Boston, MA 01451, USA; iaa551@g.harvard.edu; 7Department of Medical Laboratory Science, Kenyatta University, Nairobi 00609, Kenya; ggachara@gmail.com; 8Department of Nephrology and Hemodialysis Center, Bahre Teaching Hospital and Faculty of Medicine, International University of Africa, Khartoum 12223, Sudan; aalasmart@hotmail.com; 9Laboratory of Biological Engineering, Faculty of Science and Technology, Sultan Moulay Slimane University, Mghilla Campus, BP 523, Beni Mellal 23000, Morocco; youssef.razouqi@uit.ac.ma; 10Mogadishu Somali Turkish Training and Research Hospital, Mogadishu 2526, Somalia; m.qadar59@gmail.com; 11Department of Infection Control, Al Jawad Hospital, Khartoum 11111, Sudan; zhlabwjaj@gmail.com; 12Lagos State Ministry of Health, Ikeja 100271, Lagos, Nigeria; drtundeajayi@gmail.com; 13Pharmacovigilance and Safety Monitoring Division, Rwanda Food and Drugs Authority, Kigali P.O. Box 1948, Rwanda; ntirenganyal1@gmail.com; 14Department of Biochemistry, Lagos State University, Ojo 102101, Lagos, Nigeria; gbemisola.saibu@lasu.edu.ng; 15Walter Sisulu University, Mthatha Campus, Mthatha 5117, South Africa; 16Department of Medicine, University of Ibadan, Ibadan 200132, Oyo State, Nigeria; tundesalako@nimr.gov.ng; 17Office of the Director General, Nigerian Institute of Medical Research, Yaba, Lagos 101212, Lagos, Nigeria; 18Kwara State COVID-19 Technical Working Group, Ilorin 240241, Kwara State, Nigeria; 19Department of Community Health and Primary Health Care, Lagos State University College of Medicine, Ikeja 100271, Lagos, Nigeria; loladewright@yahoo.com; 20Lagos State University Teaching Hospital, Ikeja 100271, Lagos, Nigeria; 21Emergency Centre for Transboundary Animal Diseases (ECTAD), Food and Agricultural Organization of the United Nations (FAO), Nairobi 00100, Kenya; folorunso.fasina@fao.org; 22Department of Veterinary Tropical Diseases, University of Pretoria, Pretoria 0182, South Africa; 23Infection Control Unit, Zagazig University Hospitals, Zagazig 44511, Egypt; rashamosbah6@gmail.com; 24Faculty of Oral and Dental Medicine, Ahram Canadian University, Giza 12578, Egypt

**Keywords:** vaccine adverse events, SARS-CoV-2, COVID-19 vaccine, public health, vaccination, adverse events following immunisation (AEFI)

## Abstract

Local, national, and international health agencies have advocated multi-pronged public health strategies to limit infections and prevent deaths. The availability of safe and effective vaccines is critical in the control of a pandemic. Several adverse events have been reported globally following reception of different vaccines, with limited or no data from Africa. This cross-sectional epidemiological study investigated adverse events following COVID-19 vaccination in Africans from April–June, 2021 using a structured online questionnaire. Out of 1200 participants recruited, a total of 80.8% (*n* = 969) respondents from 35 countries, including 22 African countries and 13 countries where Africans live in the diaspora, reported adverse events. Over half of the vaccinees were male (53.0%) and frontline healthcare workers (55.7%), respectively. A total of 15.6% (*n* = 151) reported previous exposure to SARS-CoV-2, while about one-fourth, 24.8% (*n* = 240), reported different underlying health conditions prior to vaccination. Fatal cases were 5.1% (*n* = 49), while other significant heterogenous events were reported in three categories: very common, common, and uncommon, with the latter including enlarged lymph nodes 2.4% (*n* = 23), menstrual disorder 0.5% (*n* = 5), and increased libido 0.2% (*n* = 2). The study provided useful data for concerned authorities and institutions to prepare plans that will address issues related to COVID-19 vaccines.

## 1. Introduction

As the COVID-19 pandemic progresses, multilateral agencies under the coordination of the World Health Organisation (WHO) are working round the clock to stem the tide of continuing infection and transmission, not only by supplying medical kits and equipment to various healthcare systems, but also through coordinated distribution of safe and effective vaccines. Vaccination and immunisation remain the best options for the prevention and control of diseases worldwide [1]. To date, 15 November 2021, more than 7.5 billion doses of COVID-19 vaccines have been administered among 52.1% of the global population, with an approximate of 31.2 million new doses administered daily. However, only 4.5% of the population in low-income countries have been vaccinated, with approximately 300 million doses administered for full or partial vaccination in Africa (https://ourworldindata.org/covid-vaccinations?country=OWID_WRL (accessed on 7 November 2021)). It is instructive to understand that Africa has an estimated population of 1.4 billion (amounting to only 21.4% of Africans vaccinated).

The COVID-19 Vaccines Global Access (COVAX) facility was launched on the 24 February 2021 to improve COVID-19 vaccine deliveries to Africa [2]. Ghana and Cote D’Ivoire were the first two countries to receive the WHO-approved COVISHIELD vaccine on 24 and 26 February 2021, respectively. These deliveries were followed by those for Nigeria, Angola, Democratic Republic of Congo, and Gambia on the 2 March 2021. These were followed by Rwanda on the 3 March, the same day as Kenya, Sudan, and Malawi, with Rwanda being the first country to receive both the AstraZeneca/Oxford vaccine and the Pfizer–BioNTech mRNA vaccine from the COVAX facility [2]. In addition, Benin received 144,000 COVISHIELD vaccines on the 11th March 2021.

The Oxford–AstraZeneca vaccine was the most widely used because of its ease of storage at a temperature of +2 to +8 °C and associated logistics, but it is not without its adverse events, as is observed for other COVID-19 vaccines. Generally, vaccines may produce adverse reactions due to idiosyncrasies or because the body’s immunological system is set to always recognise foreign and infectious agents, and there is the possibility of it responding physiologically through the cellular and/or humoral pathways. According to the CDC in 2021, any local or systemic health problem or side effect that occurs after vaccination or immunisation is referred to as an adverse event (ADE) following vaccination or immunisation (AEFI) [3]. Global adverse events following COVID-19 vaccination vary based on the type of vaccine, but the most common symptoms reported include fatigue, headache, muscle and joint pain, allergic skin reaction, and chills, while the most prevalent events include low-grade fever and pain or redness at the site of injection, often felt a few days after vaccination. Severe adverse events are possible, but the chances are low [4,5,6,7,8].

Nevertheless, despite the identification of serious and fatal adverse events following COVID-19 vaccination, a causal relationship has not been really established [9,10,11]. Severe adverse events following vaccination have been identified by the United States of America (USA) Vaccine Adverse Event Reporting System (VAERS). According to the report of May 2021, 2 to 5 people per million vaccinated in the USA developed a severe and rare anaphylactic reaction that occurred 30 min after vaccination. VAERS also filed 32 confirmed reports of people who developed thrombosis with thrombocytopenia syndrome (TTS) after getting the Johnson & Johnson (J&J) vaccine.

Moreover, other reports identified the occurrence of myocarditis and pericarditis in vaccinees who were vaccinated with the Pfizer–BioNTech and Moderna COVID-19 vaccines, mostly in adolescents and young adults. In addition, VAERS reported 9810 human deaths from 442 million doses of COVID-19 vaccines administered in the USA from 14 December 2020, through 15 November 2021 [12]. Although a causal link of death to COVID-19 vaccines is still being examined, important reports attributed the death that occurred to a rare, serious adverse event involving blood clots with low platelets following the J&J vaccine [12]. On the other hand, thrombotic and thromboembolic adverse events and related causes of death have been reported across European countries following AstraZeneca/COVISHIELD vaccine administration [13], including but not limited to the United Kingdom [14].

Until 14 November 2021, there are a total of 15,287 (0.07%) AEFI reported following 22,749,817 doses of COVID-19 vaccines administered in Ontario, with an overall reporting rate of 67.2 per 100,000 doses and specific rates of 56.9 per 100,000 doses administered for the Pfizer–BioNTech vaccine, 81.6 per 100,000 doses for the Moderna vaccine, and 141.9 per 100,000 doses for the AstraZeneca/COVISHIELD vaccine. Furthermore, 29 reports of deaths plausibly associated with receipt of the COVID-19 vaccine have been filed, and a causal relationship to any vaccine is still being investigated [7].

Despite the outstanding achievements in the development and rapid approval of the different vaccines, Africa is still at high risk of a likely long period of COVID-19 community spread and hospitalisation, mainly due to unwillingness of Africans to receive the vaccines [15] combined with the scarcity of vaccines, lack of infrastructure for vaccine production on the continent, and paucity of funds. Furthermore, there is a dearth of information and lack of reporting or surveillance data on adverse events following COVID-19 vaccination in Africa. Furthermore, there is need for active surveillance and public health research communication in order to provide media supports for public health researchers and improve health literacy for the populace, especially among the youths, via social media to counter fake news campaigns on adverse events following COVID-19 vaccination [16].

Hence, this public health surveillance was designed to investigate adverse events associated with COVID-19 vaccination in the African population and the management options being employed by all individuals reporting adverse events.

## 2. Materials and Methods

### 2.1. Study Design and Participants

A descriptive, cross-sectional, continent-based study was carried out to monitor adverse events following COVID-19 vaccination in Africans living in Africa and diaspora from April–June 2021. An online survey instrument was deployed for data collection from consenting participants recruited using a convenience sampling method. Inclusion criteria were being an African regardless of location, 18 years of age and above, and having received any of the COVID-19 vaccines. Those who have not received the vaccines were excluded from the survey. Participants who met the inclusion criteria were only eligible if they had access to an electronic medium with Internet capability. Physical and interviewer-based questionnaires were avoided to reduce the risk of contracting and spreading SARS-CoV-2/COVID-19. An initial probable target of ≥50–≤100 respondents per country was planned to be recruited from North, Southern, West, and East Africa. This strategy was selected because it was difficult to determine the exact number of persons among the target population per country who met all the inclusion criteria for online survey participation. Since the survey was also conducted on a willing participant basis, the number of final recruits was difficult to determine. Therefore, our data should be carefully interpreted, as it does not epitomise the entire African population.

### 2.2. Questionnaire

A structured, multiple-choice pretested questionnaire was designed in English and translated into French and Arabic by native language experts to capture the specific and logical respondents’ quantitative data in Arabic- and French-speaking countries, respectively. The questionnaire was designed based on the WHO, FDA, and UK-NHS classification of common and uncommon COVID-19 vaccination reactions, to elicit information about respondents’ socio-economic and demographic characteristics, health status, past medical history, adverse events following vaccination, and how those events were managed. The questionnaire was uploaded to Google Forms (https://forms.gle/k9KYGp7wC4JNjZeL6, Alpha Inc., Turlock, CA, USA) (Appendix A) for distribution and a one-time data collection process through various online social media platforms including WhatsApp, Facebook, Instagram, Telegram, Twitter, LinkedIn, and emails. The questionnaire was first pretested among 30 respondents in different countries before its administration by research collaborators in various countries including Egypt, Ghana, Kenya, Morocco, Namibia, Nigeria, Rwanda, South Africa, Somalia, and Sudan, among others. Continuous sharing was done on social media, with sponsored adverts on Facebook deployed as reminders to encourage participation.

### 2.3. Ethics Approval and Consent

Ethical approval for the study was obtained from various institutional review committees of Nigeria (University of Ilorin), Egypt (Ahram Canadian University, Faculty of Oral and Dental Medicine) and Kenya (Kenyatta University). Guidelines and code of ethics for human research were observed in line with the Declaration of Helsinki—Ethical Principles for Medical Research involving Human Subjects (64th World Medical Association General Assembly, Fortaleza, Brazil, October 2013). Participation in the study was voluntary, allowing any participant to quit at any stage without submitting the online form. All participants consented to the study by initially selecting an option to ‘voluntarily agree or disagree to participate in the current study’, leading to the study questionnaire or ‘finished’ page as the case may be. Data collection was anonymous, and respondents’ information was kept highly confidential.

### 2.4. Data Analyses and Statistics

The collated public health surveillance data on adverse vaccine events were retrieved from the Google form in Excel format for sorting and coding. Categorical variables were presented as frequencies and proportions using descriptive statistics. Chi-square test and Fisher’s exact test for 2 × 2 tables were used to test for statistical significance between variables and the demographic (independent) values. Variables considered included demographic characteristics such as age (categories), gender, educational attainment, community type (rural, urban, semi-urban), questionnaire items concerning adverse events, and subsequent reactions following vaccination. All the vaccinated participants were included in the adverse events’ data analyses irrespective of the number of COVID-19 vaccine doses or type received. The proportion of vaccinees who reported side events post-vaccination were calculated. Differences in severity between post-vaccination adverse events were compared to investigate the variability in different categories and between variables. The event of previous exposure to SARS-CoV-2/COVID-19 and the consequence of adverse reaction was compared to the naive population.

No adjustments were made for missing data, and all analyses used complete case analysis. *p*-values were two-sided, and analyses were carried out at 95% confidence interval using Statistical Package for the Social Sciences (SPSS) software v.22 (SPSS Inc., Chicago, IL, USA) and GraphPad Prism 9.0.0 (121) (GraphPad Software Inc., San Diego, CA, USA).

## 3. Results

This study detailed the recruitment of participants, past exposure to SARS-CoV-2, exclusion criteria, hospitalisation following vaccination, the observed adverse events, and their treatment by Africans, including recommendation for COVID-19 vaccination (Figure 1).

Out of the overall participants (*n* = 1200), a total of 19.2% (*n* = 231) were excluded, as they did not meet the study inclusion criteria or declined participation by not giving their consent, while 80.8% (*n* = 969) reported different adverse events following COVID-19 vaccination from 35 countries, including 22 African countries: Cameroon, Egypt, Ethiopia, Gambia, Ghana, Liberia, Kenya, Malawi, Morocco, Namibia, Niger, Nigeria, Rwanda, Somalia, South Africa, Tanzania, Tunisia, Uganda, Swaziland, Sudan, Zambia, and Zimbabwe (Figure 2).

Nigeria recorded the highest number of participants (33.7%), followed by Ghana (23.3%) and Kenya (9.7%), respectively, while the fewest responses were documented from participants in Somalia, Sudan, and Rwanda, recording proportions of 5.1%, 4.1%, and 2.8%, respectively (Table 1).

Participants from 13 countries (USA, UK, Saudi Arabia, Canada, UAE, Bahrain, Qatar, Brazil, China, Finland, Ireland, Philippines, and Sweden) represent Africans who live in the diaspora (Figure 3).

Different demographic profiles of Africans vaccinated against COVID-19 were investigated, with the highest proportion, 40.7%, in the young (25–34 years) population and a decrease in the vaccinated population as the age increases except for the age category 18–24 years (Table 2).

A total of 15.6% (*n* = 151) reported previous exposure to SARS-CoV-2/COVID-19. Earlier investigators opined that prior exposure may alter the response of participants to COVID-19 vaccination. About a quarter, 240/969 (24.8%), of vaccinated Africans reported different underlying conditions prior to COVID-19 vaccination. Cardiovascular diseases such as high blood pressure and heart diseases accounted for more than one-third (38.8%), followed by asthma (17.9%) and diabetes (12.9%), while glaucoma and arthritis at 0.2% each were the least reported (Table 3).

The study showed a high distribution and administration of adenovector vaccines in Africa compared to other vaccines, constituting 79% of the total, with mRNA, inactivated whole virus, and live attenuated vaccines representing 10.8%, 6.6%, and 0.2%, respectively. Oxford–AstraZeneca remains the most (77.8%) administered COVID-19 vaccine across different countries in Africa (Table 4).

This study also compared adverse events in adenoviral vaccines vs. others for COVID-19 vaccination in Africa based on the different types of vaccine production technologies of adenovector (Oxford–AstraZeneca, Johnson & Johnson, Sputnik V), mRNA (Moderna, Pfizer–BioNTech, and CureVac), inactivated vaccine (Sinopharm, SinoVac/CoronaVac, and Covaxin), and live attenuated vaccine (Covi Vac). For bleeding, the highest incidence of adverse events following vaccination was reported in participants who took inactivated vaccine, at 6/57 (10.5%), followed by mRNA of 5/98 (5.1%) while adenovector accounted for the lowest, at 37/733 (5.0%). Generally, adenovector vaccines were reported to account for the lowest incidence of adverse events compared to other classes of vaccines reported on in this study (Table 5).

These data showcase incidents of serious, rare, and fatal cases following COVID-19 vaccination, including bleeding/unusual weakness at 5.3% (*n* = 51), reported number of deaths at 5.1% (*n* = 49), convulsion at 4.3% (*n* = 42), breathing difficulty at 2.7% (*n* = 26), and hearing/vision problems at 2.2% (*n* = 21). It is probable that these may have been missed in clinical trials despite the willingness of Africans 35.8% (*n* = 347) to partake in clinical trials (Table 6).

We have determined some heterogenous adverse events following COVID-19 vaccines’ administration in Africa. These heterogenous adverse events among Africans were statistically significant and can be reported in three categories: uncommon, common, and very common signs. Reported uncommon signs include feeling dizzy at 11.9% (*n* = 116), abdominal pain at 3.3% (*n* = 32), itchy skin or rash at 2.9% (*n* = 28), enlarged lymph nodes at 2.4% (*n* = 23), and menstrual disorder at 0.5% (*n* = 5). Out of the common adverse events, fever at 33% (*n* = 320), injection site swelling, redness, or lump at 18.2% (*n* = 176), and influenza-like symptoms at 12.1% (*n* = 117) were the most reported, while fatigue at 40% (*n* = 388), tenderness at 39.2% (*n* = 380), and headache at 37.5% (*n* = 363) represent the topmost very common signs (Table 7).

A large proportion of Africans with adverse events following COVID-19 vaccination, 37% (*n* = 361), managed and treated the reactions at home without any visit to the hospital, while 1% (*n* = 7) sought the option of traditional remedy (Figure 4).

Study participants were vaccinated for COVID-19 significantly based on the recommendations given by healthcare workers (23.8%), various levels of governments (17.9%), personal research/investigation (15.0%), public health organisations/institutions (14.3%), and scientists (13.5%). These sources are about 5–10 times greater enablers of COVID-19 vaccination compared to the recommendations received from religious and community leaders (2.7%), social media (2.7%), not-for-profit organisations (NGOs) (0.8%), and celebrities (0.3%) (Figure 5).

## 4. Discussion

This is a report of the first pan-African findings on rare and fatal cases following COVID-19 vaccination among Africans, using the common vaccines administered in the continent. It also detailed past exposure to SARS-CoV-2 and hospitalisation following vaccination, how adverse events were treated by Africans, and how they were recommended for COVID-19 vaccination (Figure 1). The major strengths of this research include the fact that it is a population-based and geographically diverse study over time [17]. Surveillance for adverse events following vaccination is crucial to ensure safety, maintain trust, and guide policy-makers. This is further corroborated by several studies reported elsewhere, including in Asia [18,19,20], America [12,16], and Europe [21,22].

In this study, respondents from Nigeria, Ghana, and Kenya contributed more to the survey (Table 1); whether this was due to willingness of the respondents to complete the questionnaire, the intensity of administration of the questionnaire, ability to access the questionnaires, or the effect of population distributions is unclear. However, the reasons for low participation in online surveys in Africa are obvious and multifaceted, including but not limited to the lack of power supply, poor access to or irregular Internet and electronic gadgets, lack of motivation, perception of questionnaires as too burdensome or intrusive, unexplained context of the objectives, or ethics-related issues.

Different demographic profiles of Africans vaccinated against COVID-19 were investigated, including the young population, who are at higher risk of blood clots [23] and other adverse events. We observed a decrease in the vaccinated population as age increases except for the age category 18–24 years. This trend agreed with previous reports that youths are more willing to partake in vaccination compared to the older population [19,24]. Furthermore, previous work reported the largest number of doses (5,124,940) in vaccinees aged 18 to 49 years [17]. It should, however, be noted that online questionnaire administration may also facilitate more inclusion of the youth group.

More than half of the vaccinated population are male (53.0%) (Table 2), and this observation is similar to previous findings, though no significant statistical difference was observed in AEFI reported based on gender, implying no correlation between gender and reactogenicity [20]. The proportion of frontline healthcare workers that participated in the study was high (55.7%), an unsurprising observation. Is it important to prioritise vaccination and proper surveillance of AEFI among healthcare workers in order to reduce occupational hazards and the burden of hospitalisation associated with SARS-CoV-2/COVID-19 infection.

In this study, a total of 15.6% (*n* = 151) reported previous exposure to SARS-CoV-2/COVID-19, and earlier investigators opined that prior exposure may alter the response of participants to COVID-19 vaccination. Further studies are needed to unravel the effect and mechanism of immunological response of past exposure to SARS-CoV-2/COVID-19 and vaccination, especially among the African populace. Although there are burgeoning data on the mystery and mechanistic roles surrounding the interference of these underlying health conditions in the general population, including among Africans [25], they remain potential predictors of various adverse events following COVID-19 vaccination.

The data reported here showcase incidents of serious rare and fatal cases following COVID-19 vaccination. It is probable that some of these may have been missed in clinical trials; only 35.8% (*n* = 347) of Africans showed willingness to partake in clinical trials (Table 6). The serious, rare, and fatal cases reported amongst Africans agree with the adverse events and deaths reported by the European Medicines Agency [21], including the fatal cases, along with the blood, ear, eye, and respiratory disorders following COVID-19 vaccination.

Although the cause of death was reported as COVID-19 vaccination complication, as stated and confirmed by the deceased’s loved ones, families, friends, and healthcare workers/caregivers, coupled with the fact that a high proportion of the vaccinees reported had underlying/co-morbid conditions, this does not sufficiently prove that the vaccines caused the adverse events. Even though this study classified adverse events following COVID-19 vaccination as serious in accordance with the WHO definition that an adverse events following vaccination is considered serious if it is life-threatening, results in hospitalisation or disability/incapacity, or leads to fatality, there is need for further investigation using the WHO causality assessment algorithm to classify adverse events following immunisation/vaccination as ‘consistent causal association’, ‘inconsistent causal association’, ‘indeterminate’, or ‘not-classifiable’ in relation to COVID-19 vaccination among Africans [26,27]. The WHO causality assessment algorithm will further provide information on vaccine safety based on the causal link between vaccines and serious adverse events following COVID-19 vaccination.

Although some of the adverse events were reported to resolve within a few days after vaccination, in actual fact, they may be the reactions of the immune system shortly after vaccination (also known as reactogenicity) [22,28]. The CDC recommends that individuals having severe allergic reactions immediately (within 4 h) or some days after administration of the vaccine should refrain from getting a second shot of the type of vaccine that produced the event [29]. Generally, the WHO, CDC, major international organisations, and experts in vaccinology agree that the efficacy of the current COVID-19 vaccines at reducing the spread of SARS-CoV-2, disease complications, and deaths far outweighs the likely risk of adverse events.

We have determined some heterogenous adverse events to have followed COVID-19 vaccine’ administration in Africa. These heterogenous adverse events among Africans were statistically significant and can be reported in three categories: uncommon, common, and very common, as reported in the results section. These adverse events among Africans reported here align with similar ones reported from other continents, including dizziness, fever, and tenderness at the injection site, while abdominal pain and menstrual disorder were less reported in other populations [19,21,22].

Nonetheless, transient local inflammation signalling neutrophils and antigen-presenting cells to the site of injection is expected after intramuscular administration of lipid-nanoparticle-formulated mRNA vaccines. According to the CDC, following the administration of the first dose of the COVID-19 vaccine, if an itch or swollen or painful rash is observed in a person, said person(s) should be treated with antihistamine or acetaminophen. If fatigue or pain is observed, treatment is equally recommended before such candidates proceed for the second shot of the vaccine based on availability to affirm complete protection [29]. In all cases, it becomes necessary to critically evaluate patients’ previous medical histories and vaccine-associated allergies in detail; it is also important to monitor vaccinated persons for at least 30 min following COVID-19 vaccine administration to ensure that no immediate untoward events are observed.

In this study, adverse events were managed and treated at home without any visit to the hospital (37%; *n* = 361) or with the use of traditional remedy (1%; *n* = 7). These observations agree with the previous review [22] that only a limited proportion of patients who experienced serious adverse events elsewhere considered treatment. This may be a reflection of the following: (1) the under-resourced situation of most healthcare facilities in Africa; (2) the lack of healthcare service delivery where it is most-needed, particularly, among the poor, and (3) the costs associated with seeking healthcare in Africa, including the lack of health insurance for the majority of the populations. In addition, previous experiences of people seeking hospitalisation including wrong or unfavourable evaluations and diagnoses by physicians or healthcare organisations, the low perceived need to seek medical care due to views that the illnesses or symptoms will improve over time, and time constraints on visiting hospitals may also be major constraints [30].

The study participants were vaccinated for COVID-19, based on various recommendations or knowledge channels (Figure 5), with the five top sources being 5–10 times greater enablers of COVID-19 vaccination compared to the least used sources (religious and community leaders (2.7%), social media (2.7%), not-for-profit organisations (NGOs) (0.8%), and celebrities (0.3%)); it is recommended that for society to benefit optimally from COVID-19 vaccine coverage, the earlier proposal of Anjorin et al. [15] on the use of a multi-channel vaccination campaign strategy is necessary for excellent vaccine uptakes.

In the new world, a reporting system known as the Vaccine Adverse Event Reporting System (VAERS) has been introduced by the CDC and US Food and Drug Administration (FDA) to record any adverse events known to occur in the future after vaccination. The system also proffers invaluable information to vaccinologists to guarantee safety [31]. Whether such technologies will reach wide adoption in Africa is doubtful. However, select African countries are utilising similar technologies. This includes, for instance, the Med Safety App, originally developed by the WEB-Recognising Adverse Drug Reactions (WEB-RADR) project, which was aimed to enable self-reporting suspected adverse events following vaccines or drugs for proper monitoring. In Nigeria, for instance, such monitoring is conducted by the National Agency for Food and Drug Administration and Control (NAFDAC) (https://www.nafdac.gov.ng/wp-content/uploads/Publications/Others/Events_PDF/how-to-download-the-med-safety-app-1-1.pdf (accessed on 30 November 2021)), and in South Africa, the tool is equally applied (Med Safety App—SAHPRA (https://medsafety.sahpra.org.za (accessed on 30 November 2021))). Therefore, there is a need to promote the utilisation of easy-to-access and report tools such as the above and encourage other African countries that are yet to voluntarily adopt and utilise such available reporting systems to join the league of adoptees for proper documentation and future informed decision-making.

While COVID-19 vaccines are undoubtedly beneficial, the safety concerns, including those adverse events identified in this work, hospitalisation, fatalities, and excess economic spending on health are relevant and should not be ignored [32]. Nevertheless, the benefits of being vaccinated against COVID-19 for perceived disease risks outweigh these identified demerits, and a decision on accepting the vaccine should be strengthened on this basis [33]. Given that it is difficult to determine what degree of protection against COVID-19 disease is attributable to the vaccine’s effects in the body, the risk perceptions of COVID-19 are often measured as the perceived likelihood of contracting the disease and the perceived severity of the symptoms. These perceptions also have an emotional dimension, including fear and worry components [34]. In Catalonia, based on a previous model, the benefit/cost ratio of COVID-19 vaccination was estimated at 3.4 from a social perspective and 1.4 from a health system perspective, and such benefits are associated with the monetisation of the reduction in mortality and cases with sequelae, intervention, and reduction in the use of resources [35].

We have evaluated the adverse events following the use of COVID-19 vaccines in Africa and report our findings. In this study, a number of proposed pathogenic mechanisms may have been responsible, through which the vaccines may have produced the observed adverse events, including the following: Firstly, the vaccine-induced immune thrombotic thrombocytopenia, which trigger platelet activation. Through the vaccine adenoviral non-replicating vectors (AVV), the COVID-19 vaccines may (i) spread rapidly into the blood stream, (ii) promote the early production of high levels of IL-6, (iii) interact with erythrocytes, platelets, mast cells, and endothelia, (iv) favour the presence of extracellular DNA at the site of injection, and (v) activate platelets and mast cells to release PF4 and heparin [36]. Secondly, the mechanism of anaphylaxis may also be IgE-mediated, with polyethylene glycol as the inciting antigen. However, other unknown complement-mediated mechanisms may be possible in individuals without a previous history of allergy [37].

However, our study is subjected to some limitations. For one, since the study is based on a willingness to participate, it became difficult to obtain proportional representation per country or to have all the countries enrolled based on density of vaccination per country or other logical considerations. This may have affected the statistically empirical determination of expected number of participants per country, thus skewing the analyses, including for the outcomes that are less frequent. It should be noted that some variables and predictors of interest were not captured in this preliminary report, perhaps due to the intrusive nature of the associated questions, which may discourage the participants from cooperating with filling the questionnaires.

## 5. Conclusions

We hereby report adverse events following COVID-19 vaccination from a total of 80.8% (*n* = 969) Africans in 35 different countries. Previous exposure to SARS-CoV-2/COVID-19 was reported by 15.6% (*n* = 151) participants with various underlying diseases. Oxford–AstraZeneca remains the most (77.8%) administered COVID-19 vaccine across African countries. The most worrisome, rare, and fatal cases include bleeding/unusual weakness at 5.3% (*n* = 51), reported number of deaths at 5.1% (*n* = 49), convulsion at 4.3% (*n* = 42), breathing difficulty at 2.7% (*n* = 26), and hearing/vision problems at 2.2% (*n* = 21) while heterogenous adverse events were reported in three categories: very common, common, and uncommon, with the latter including abdominal pain 3.3% (*n* = 32), enlarged lymph nodes 2.4% (*n* = 23), and menstrual disorder 0.5% (*n* = 5). Perhaps an advanced, government-based, more robust real-time online data-capturing system for reporting continuous sentinel AEFI surveillance, instituted in different African countries, from which future data may be mined for analysis, may be necessary. Such a database may be managed by the Africa Union or its agency, the Africa Centres for Disease Control and Prevention, on behalf of the member states, similar to databases that exist in other continents.

## Figures and Tables

**Figure 1 vaccines-10-00546-f001:**
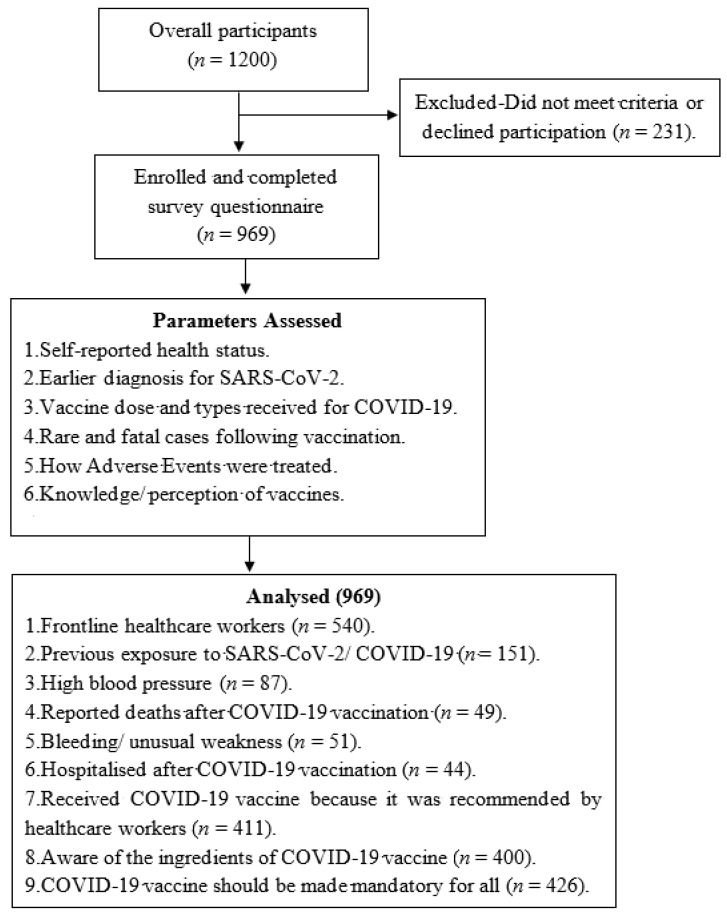
Flow chart of participants experiencing adverse events following COVID-19 Vaccination in Africa.

**Figure 2 vaccines-10-00546-f002:**
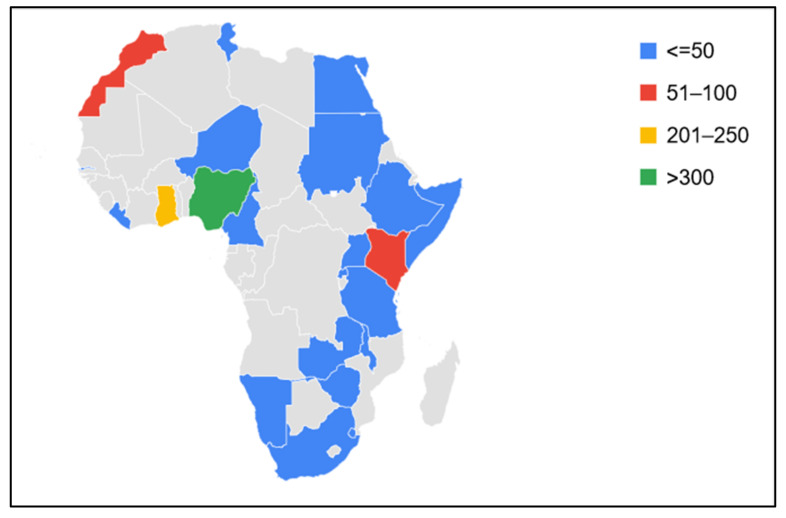
Distribution of respondents to questions on adverse events following COVID-19 vaccination in African countries.

**Figure 3 vaccines-10-00546-f003:**
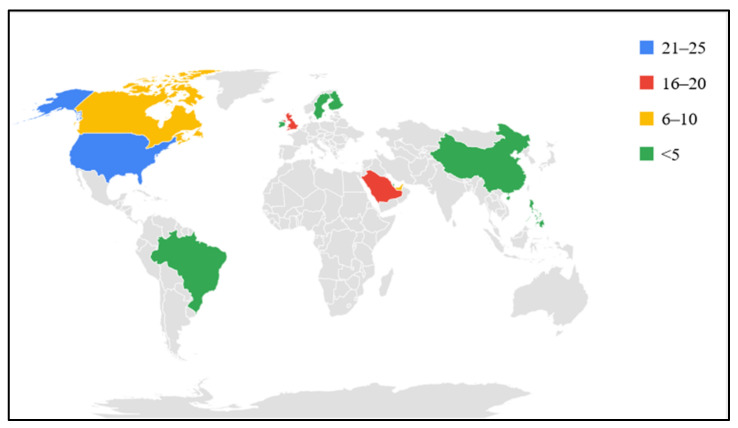
Distribution of some Africans vaccinated against COVID-19 living in the diaspora.

**Figure 4 vaccines-10-00546-f004:**
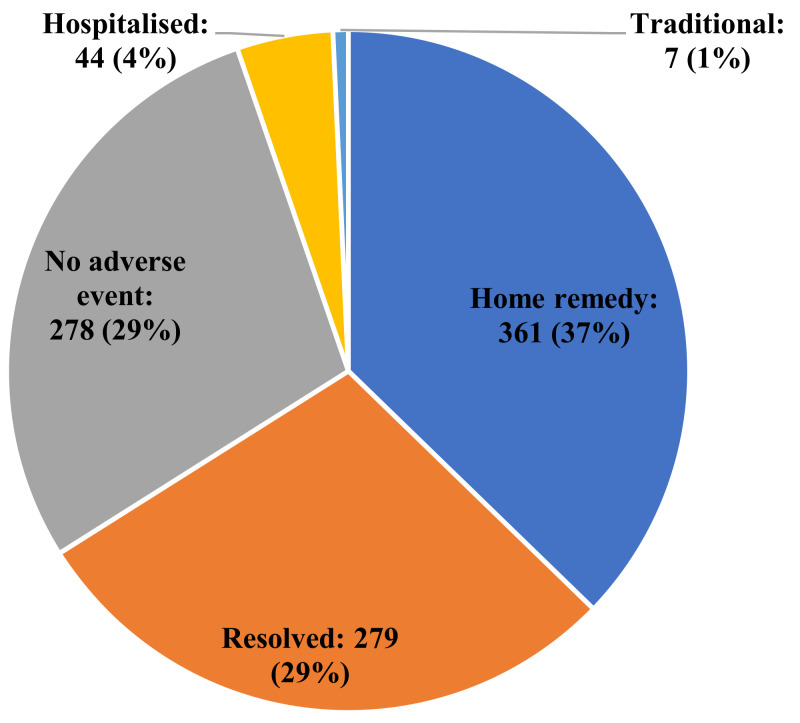
How adverse events of COVID-19 vaccination were treated among Africans.

**Figure 5 vaccines-10-00546-f005:**
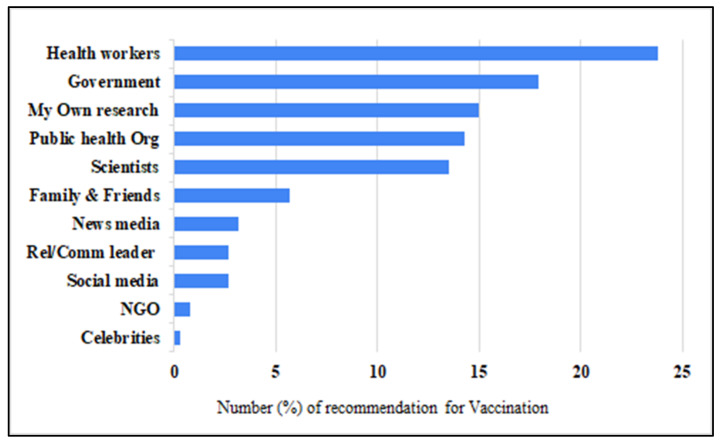
Distribution of recommendations that enhanced COVID-19 vaccination among Africans.

**Table 1 vaccines-10-00546-t001:** Distribution in some African countries based on the number of willing participants.

Country	Frequency	Proportion (%)
Nigeria	327	33.7
Ghana	226	23.3
Kenya	94	9.7
Diaspora	83	8.6
Morocco	52	5.4
Egypt	50	5.2
Somalia	49	5.1
Sudan	40	4.1
Rwanda	27	2.8

**Table 2 vaccines-10-00546-t002:** Demographic distribution of Africans vaccinated for COVID-19.

Category	Frequency	Percentage
**Age**		
18–24	84	8.7
25–34	394	40.7
35–44	268	27.7
45–54	123	12.7
55–64	69	7.1
>65	31	3.2
**Gender**		
Male	514	53.0
Female	455	47.0
Education		
Tertiary	804	83.0
Secondary	49	5.1
Primary	10	1.0
Others	89	9.2
None	17	1.8
**Occupation**		
Frontline Healthcare Workers	540	55.7
Frontline Non-healthcare Workers	127	13.1
**Others**	302	31.2
Community		
Urban	727	75.0
Semi-urban	149	15.4
Rural	93	9.6

**Table 3 vaccines-10-00546-t003:** African respondents earlier diagnosed positive for SARS-CoV-2 and underlying conditions before COVID-19 vaccination.

	Frequency (*n* = 969)	Proportion (%)
**Earlier diagnosed positive**		
No	799	82.5
Yes	151	15.6
Missing	19	2.0
**Underlying conditions**		
No	729	75.2
Yes	240	24.8
CVD	93	38.8
Asthma	43	17.9
Diabetes	31	12.9
Obesity	26	10.8
Infectious diseases	14	5.8
GIT Diseases	10	4.2
Haematological disorder	9	3.8
Cancer	5	2.1
Arthritis	2	0.8
Glaucoma	2	0.8
Others	5	2.1

**Table 4 vaccines-10-00546-t004:** Vaccine dose and types administered for COVID-19 in Africa.

	Frequency *n* = 969	Proportion %
**Vaccine dose**		
First dose	811	83.7
Complete dose	158	16.3
**Vaccine type**		
Adenovector	766	79
mRNA	105	10.8
Inactivated whole virus	64	6.6
Live attenuated vaccine	2	0.2
Others	32	3.3
**Vaccine Brand**		
Oxford–AstraZeneca	754	77.8
Johnson & Johnson	5	0.5
Covaxin	9	0.9
Sinopharm-BBIBP	44	4.5
Moderna	17	1.8
Pfizer–BioNTech	88	9.1
CoronaVac	6	0.6
Sputnik V	7	0.7
Sinopharm-WIBP	5	0.5
Covi Vac	2	0.2
Others	32	3.3

**Table 5 vaccines-10-00546-t005:** Comparison of adverse events in adenoviral vaccines vs. others for COVID-19 in Africa.

Bleeding	*n* = 918	AEFV	%	*p*-Value
Adenovector	733	37	5	0.33
mRNA	98	5	5.1	
Inactivated vaccine	57	6	10.5	
Live attenuated	2	0	0	
Others	28	3	10.7	
**Seizure**				
Adenovector	728	31	4.3	0.10
mRNA	102	3	2.9	
Inactivated vaccine	60	4	6.7	
Live attenuated	2	0	0	
Others	28	4	14.3	
**Breathing difficulty**				
Adenovector	750	19	2.5	0.90
mRNA	100	3	3	
Inactivated vaccine	61	2	3.3	
Live attenuated	2	0	0	
Others	30	5	16.7	
**Hearing/Vision**				
Adenovector	757	11	1.5	0.09
mRNA	101	3	2.9	
Inactivated vaccine	60	3	5	
Live attenuated	2	0	0	
Others	28	4	14.3	
**Severe allergic reaction**				
Adenovector	752	16	2.1	0.35
mRNA	102	3	2.9	
Inactivated vaccine	60	3	5	
Live attenuated	2	0	0	
Others	28	4	14.3	

**Table 6 vaccines-10-00546-t006:** Rare and fatal cases following COVID-19 vaccination among Africans.

	Frequency (*n* = 969)	Percent (%)	*p*-Value
**Bleeding/Unusual weakness**			
No	918	94.7	<0.0001
Yes	51	5.3	
**Died after vaccination**			
No	920	94.9	<0.0001
* Yes	49	5.1	
**Seizure (convulsion) or high fever after hours or a few days**			
No	927	95.7	<0.0001
Yes	42	4.3	
**Breathing difficulty**			
No	943	97.3	<0.0001
Yes	26	2.7	
**Hearing/Vision problem**			
No	948	97.8	<0.0001
Yes	21	2.2	
**Clinical trial**			
No	622	64.2	<0.0001
Yes	347	35.8	

* The deaths were accounted for by the healthcare workers that attended to the vaccinees with adverse events leading to deaths or the family members of the dead persons.

**Table 7 vaccines-10-00546-t007:** Adverse events following COVID-19 vaccination among Africans.

	Frequency (*n* = 969)	Percentage	*p*-Value
I experienced uncommon signs including:			<0.0001
None	651	67.1	
Feeling dizzy	116	11.9	
Decreased appetite	62	6.4	
Excessive sweating	41	4.2	
Abdominal pain	32	3.3	
Itchy skin or rash	28	2.9	
Enlarged lymph nodes	23	2.4	
Menstrual disorder	5	0.5	
Hunger	4	0.4	
Increased libido	2	0.2	
I experienced common signs including:			
None	508	52.4	<0.0001
Fever	320	33.0	
Swelling, redness or a lump at the injection site	176	18.2	
Flu-like symptoms such as high temperature, sore throat, runny nose, cough and chills	117	12.1	
Being sick (vomiting)	44	4.5	
Diarrhoea	20	2.1	
Heaviness of the head	2	0.2	
Bone ache	1	0.1	
Lymph node enlargement	1	0.1	
I experienced very common signs including:			<0.0001
None	220	22.7	
Feeling tired/fatigued	388	40.0	
Tenderness, pain, warmth, itching or bruising where the injection was given	380	39.2	
Headache	363	37.5	
Generally feeling unwell	339	34.9	
Chills or feeling feverish	293	30.2	
Joint pain/muscle ache	269	27.8	
Feeling sick/nausea	115	11.9	
Deep sleep	5	0.5	
Lymph in armpits	3	0.3	
Mouth sores	1	0.1	
Boil	1	0.1	
Experienced lower sex drive	1	0.1	
Diarrhoea	1	0.1	
Ear pain	1	0.1	
Chest pain	1	0.1	
Vomiting	1	0.1	
Blood (red) spot on left eye	1	0.1	
Tender swollen tongue, loss of taste and appetite	1	0.1	
Insomnia	1	0.1	
Dry cough	1	0.1	
Rhinitis	1	0.1	
Numbness at neck and hand after 2nd dose for one night	1	0.1	

## Data Availability

Relevant data have been included in this manuscript.

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
