# Peer review of "Public Health Surveillance for Adverse Events Following COVID-19 Vaccination in Africa"

_vaccines, 2022, doi:10.3390/vaccines10040546_

Round 1

Reviewer 1 Report

This is a very interesting paper that contributes to increasing our knowledge on COVID- 19 vaccination. It reports epidemiological data that are well - presented and structured. Limitations of the study are well highlighted by Authors themselves. I suggest Authors to spend some words on  risk - benefits analysis of COVID vaccination, just to clarify the undoubted benefits of vaccination. 

Author Response

Response: Thank you for the positive comments from the reviewer. We have now addressed the comment that he raised in the review by adding the following comment below to the discussion.

‘While COVID-19 vaccine is undoubtedly beneficial, the safety concerns including those adverse events identified in this work, hospitalization, fatalities and excess economic spending on health are relevant and should not be ignored (Shaker, Abrams and Greenhawt, 2021). Nevertheless, the benefits of being vaccinated against COVID-19 perceived disease risks outweigh these identified demerits, and a decision on accepting the vaccine should be strengthened on this basis (Karlsson et al., 2021). Given that it is difficult to determine what degree of protection against COVID-19 disease is attributable to the vaccine events in the body, the risk perceptions on COVID-19 are often measured as the perceived likelihood of contracting the disease and the perceived severity of the symptoms. These perceptions also have an emotional dimension, as well as fear and worry components (Loewenstein et al., 2001). In Catalonia, based on previous model, the benefit/cost ratio of COVID-19 vaccination was estimated at 3.4 from a social perspective and 1.4 from a health system perspective, and such benefits are associated with the monetization of the reduction in mortality and cases with sequelae, the intervention, and reduction in the use of resources (Lopez et al, 2022)’

The additional references are now added as below:

López, F.; Català, M.; Prats, C.; Estrada, O.; Oliva, I.; Prat, N.; Isnard, M.; Vallès, R.; Vilar, M.; Clotet, B.; Argimon, J.M.; Aran, A.; Ara, J. A Cost–Benefit Analysis of COVID-19 Vaccination in Catalonia. Vaccines 2022, 10, 59. https://doi.org/10.3390/vaccines10010059.

Karlsson, C.K.; Soveri, A.; Lewandowsky, S.; Karlsson, L.; Karlsson, H.; Nolvi, S.; Karukivi, M.; Lindfelt, M.; Antfolk, J. Fearing the disease or the vaccine: The case of COVID-19. Personality and Individual Differences, 2021; 172, 110590, https://doi.org/10.1016/j.paid.2020.110590.

Loewenstein, G.F.; Weber, E.U.; Hsee, C.K.; Welch, N. Risk as feelings. Psychological Bulletin, 2001; 127(2), 267-286, 10.1037/0033-2909.127.2.267

Shaker, M.; Abrams, E.M.; Greenhawt, M. A Cost-Effectiveness Evaluation of Hospitalizations, Fatalities, and Economic Outcomes Associated with Universal Versus Anaphylaxis Risk-Stratified COVID-19 Vaccination Strategies. The journal of allergy and clinical immunology. In practice, 2021; 9(7), 2658–2668.e3. https://doi.org/10.1016/j.jaip.2021.02.054.

Reviewer 2 Report

Q1. Methods. Please better explain the sample size calculation.

Q2. Line 130. Correct the typo

Q3. Table 6 should be reported after the sample description at the beginning of results paragraph

Q4. I suggest to split results and discussion in two paragraphs 

Q5. I suggest to perform a sub-analysis comparing the AEFIs in adenoviral vaccines vs. others.

Q6. You do not focus on causal correlation. You should add a paragraph on "causality assessment" and its implication on your results. I suggest this study to use as hint doi: 10.3390/vaccines7040140

Q7. You should add few sentences on consequence of adverse reaction communication by traditional and social media and how it can influence (positively or negatively) the success of vaccination campaign. I suggest doi: 10.7416/ai.2022.2499.

Author Response

Q1. Methods. Please better explain the sample size calculation.

Response: The following statement has now been included in lines 142 – 147: ‘An initial probable target of ≥ 50 – ≤ 100 respondents per country was planned to be recruited from North, Southern, West, and East Africa. This strategy was selected because it was difficult to determine the exact number of target population per country who met all the inclusion criteria for an online survey participation. Since the survey was also on a willing participant basis, the number of final recruits was difficult to determine. Therefore, our data should be carefully interpreted as it does not epitomise the entire African population.

Q2. Line 130. Correct the typo

Response: The observed typographical error was corrected.

Q3. Table 6 should be reported after the sample description at the beginning of results paragraph

Response: This has been adjusted with Table 6 reported after the sample description at the beginning of results paragraph.

Q4. I suggest to split results and discussion in two paragraphs 

Response: Result and discussion are now split and are presented in the manuscript differently.

Q5. I suggest to perform a sub-analysis comparing the AEFIs in adenoviral vaccines vs. others.

Response: Sub-analysis comparing the proportion of adenoviral vaccines vs. others has been added to compare their distributions (Table 4 and Table 5): The study showed a high distribution and administration of adenovector vaccines in Africa compared to other vaccines including mRNA, inactivated whole virus and live attenuated vaccines having 79%, 10.8%, 6.6% and 0.2% respectively.

This study also compared adverse events in adenoviral vaccines vs. others for COVID-19 vaccination in Africa based on the different types of vaccine production technologies of adenovector (Oxford-AstraZeneca, Johnson & Johnson, Sputnik V), mRNA (Moderna, Pfizer-BioNTech, and CureVac), inactivated vaccine (Sinopharm, SinoVac/ CoronaVac and Covaxin) and live attenuated vaccine (Covi Vac) (Table 5). For bleeding, the highest adverse events following vaccination was reported in 6/57 (10.5%) of participants who took inactivated vaccine, followed by mRNA of 5/98 (5.1%) while adenovector accounted for the lowest 37/733 (5.0%). Generally, adenovector vaccines were reported to account for the lowest adverse events compared to other class of vaccines reported in this study.

Q6. You do not focus on causal correlation. You should add a paragraph on "causality assessment" and its implication on your results. I suggest this study to use as hint doi: 10.3390/vaccines7040140

Response: Thank you for this great observation. We have now added the suggestion as part of our recommendation and future study as follows:

Although this study classified adverse events following COVID-19 vaccination as serious in accordance with the WHO definition that an adverse events following vaccination is considered serious if it is life-threatening; results in hospitalisation, disability/incapacity or leads to fatality, there is need for further investigation using the WHO causality assessment algorithm to classify adverse events following immunisation/ vaccination as ‘consistent causal association’, ‘inconsistent causal association’, ‘indeterminate’ or ‘not-classifiable’ on serious COVID-19 vaccination among Africans (WHO, 2018; Stefanizzi, 2019). The WHO causality assessment algorithm will further provide information on vaccine safety based on the causal link between vaccine and serious adverse events following COVID-19 vaccination.

References:

WHO. Causality Assessment of an Adverse Event Following Immunization (AEFI). User Manual for the Revised WHO Classification, January 2018. Available online: https://apps.who.int/iris/bitstream/handle/10665/259959/9789241513654-eng.pdf?sequence=1&isAllowed=y.

Stefanizzi P, Stella P, Ancona D, Malcangi KN, Bianchi FP, De Nitto S, Ferorelli D, Germinario CA, Tafuri S. Adverse Events Following Measles-Mumps-Rubella-Varicella Vaccination and the Case of Seizures: A Post Marketing Active Surveillance in Puglia Italian Region, 2017-2018. Vaccines (Basel). 2019;7(4):140. doi: 10.3390/vaccines7040140.

Q7. You should add few sentences on consequence of adverse reaction communication by traditional and social media and how it can influence (positively or negatively) the success of vaccination campaign. I suggest doi: 10.7416/ai.2022.2499.

Response: Added as below:

Also, there is need for active surveillance and public health research communication with the provision of media supports for public health researchers, improve health literacy for the populace especially among the teaming youths via social media to counter fake news campaign on adverse events following COVID-19 vaccination (Bianchi ad Tafuri, 2022).

Reference:

Bianchi FP, Tafuri S. A public health perspective on the responsibility of mass media for the outcome of the anti-COVID-19 vaccination campaign: the AstraZeneca case. Ann Ig. 2022 Feb 3. doi: 10.7416/ai.2022.2499.

Reviewer 3 Report

This is an interesting article focused on the adverse effects of covid 19 vaccines.

There was a 5% mortality, what were the causes of death?

In relation to the adverse effects of hearing and sight, what were these effects attributed to the vaccines?

Is it important that the authors mention which are the proposed pathogenic mechanisms, through which the vaccines produce adverse effects?

It is important to separate the results section and then describe  the discussion.

Author Response

Responses:

Q1: This is an interesting article focused on the adverse effects of covid 19 vaccines.

Response: Thank you for this encouragement.

Q2: There was a 5% mortality, what were the causes of death?

Response: We have now briefly added the cause of death as: the cause of death was reported as COVID-19 vaccination complication as stated and confirmed by the deceased loved ones, families, friends and healthcare workers/ caregivers. Also, a high proportion of the vaccinees reported had underlying/ co-morbid conditions.

Q3: In relation to the adverse effects of hearing and sight, what were these effects attributed to the vaccines?

Response: Thank you for the comment. It should be understood that these are self-responses by respondents or their health attendants, and we did not differentiate between temporary and permanent effect. We are however aware that some medication and biologicals may produce temporary impairment of vision or hearing, for instance, chloroquine. Santovito and Pinna has reported reduced visual acuity due to Pfizer-Biontech COVID-19 vaccine too. Santovito, L. S., & Pinna, G. (2021). Acute reduction of visual acuity and visual field after Pfizer-BioNTech COVID-19 vaccine 2nd dose: a case report. Inflammation research: official journal of the European Histamine Research Society ... [et al.], 70(9), 931–933. https://doi.org/10.1007/s00011-021-01476-9

Q4: Is it important that the authors mention which are the proposed pathogenic mechanisms, through which the vaccines produce adverse effects?

Response: The pathogenic mechanisms through which COVID-19 vaccines may have produced the vaccine-associated adverse events may include: 1) Vaccine-Induced immune thrombotic thrombocytopenia, which trigger platelet activation. The vaccine through the adenoviral non replicating vectors (AVV) can i) spread rapidly into blood stream, ii), promote the early production of high levels of IL-6, iii) interact with erythrocytes, platelets, mast cells and endothelia, iv) favor the presence of extracellular DNA at the site of injection, v) activate platelets and mast cells to release PF4 and heparin (Azzarone et al. 2021)

2) The mechanism of anaphylaxis may also be IgE mediated, with polyethylene glycol as the inciting antigen. However, other complement-mediated mechanisms may be possible in individuals without a previous history of allergy (Edwards and Orenstein, 2022).

Azzarone B, Veneziani I, Moretta L, Maggi E. Pathogenic Mechanisms of Vaccine-Induced Immune Thrombotic Thrombocytopenia in People Receiving Anti-COVID-19 Adenoviral-Based Vaccines: A Proposal. Front Immunol. 2021 Aug 13;12:728513. doi: 10.3389/fimmu.2021.728513.

Edwards K.M.; Orenstein, W.A. COVID-19: Vaccines. Available at: https://www.uptodate.com/contents/covid-19-vaccines. Assessed 05 March 2022.

These are now added to the discussions.

Q5: It is important to separate the results section and then describe the discussion.

Response: The discussion is now separated as below.

This is a report of the first pan-African findings on rare and fatal cases following COVID-19 vaccination among Africans, using the common vaccines administered in the continent. It also detailed past exposure to SARS-CoV-2 and hospitalisation following vaccination, how adverse events were treated by Africans, and how they were recommended for COVID-19 vaccination (Figure 1). The major strengths of this research include the fact that it is a population-based, and geographically diverse study over time [16]. Surveillance for adverse events following vaccination is crucial to ensure safety, maintain trust, and guide policy-makers. This is further corroborated by several studies reported elsewhere including in Asia [17-19], America [12,16], and Europe [20,21].

In this study, different demographic profiles of Africans vaccinated against COVID-19 were investigated, including the young population who are at higher risk of blood clots [22] and other adverse events. We observed a decrease in the vaccinated population as the age increases except for the age category 18-24 years. This trend agreed with previous reports that youths are more willing to partake in vaccination compared to the older population [18,23]. Furthermore, previous work reported the largest number of doses (5,124,940) from vaccinees aged 18 to 49 years [16]. It should however be noted that online questionnaire administration may also facilitate more inclusion of the youth group.

More than half of the vaccinated population are male (53.0%) (Table 1), and this observation is similar to previous finding though no significant statistical difference was observed in AEFI reported based on gender, an implication of no correlation between gender and reactogenicity [19]. The proportion of frontline healthcare workers that participated in the study was high (55.7%), an unsurprising observation. Is it important to prioritise vaccination and proper surveillance of AEFI among healthcare workers in order to reduce occupational hazard and the burden of hospitalisation associated with SARS-CoV-2/ COVID-19 infection.

In this study, a total of 15.6% (n=151) reported previous exposure to SARS-CoV-2/ COVID-19, and earlier investigators opined that prior exposure may alter the response of participants to COVID-19 vaccination. Further studies are needed to unravel the effect and mechanism of immunological response of past exposure to SARS-CoV-2/COVID-19 and vaccination, especially among the African populace. Although there is burgeoning data on the mystery and mechanistic roles surrounding the interference of these underlying health conditions in the general population, including among Africans [24], they remain potential predictors of various adverse events following COVID-19 vaccination.

The data reported here showcased incidents of serious rare and fatal cases following COVID-19 vaccination, it is probable that some of these may have been missed out in clinical trials; only 35.8% (n=347) of Africans showed willingness to partake in clinical trials (Table 4). The serious rare and fatal cases reported amongst Africans agree with the adverse events and deaths reported by the European Medicines Agency [20], including the fatal cases, along with the blood, ear, eyes, and respiratory disorders following COVID-19 vaccination. However, it does not sufficiently prove that the vaccines caused the adverse events.

Although some of the adverse events were reported to resolve within a few days after vaccination, in actual fact, they may be the reaction of the immune system shortly after vaccination (also known as reactogenicity) [21,25]. The CDC recommends that individuals having severe allergic reactions immediately (within 4 hours) or some days after administration of the vaccine should refrain from getting a second shot of that type of vaccine that produced the event [26]. Generally, the WHO, CDC, and major international organisations and experts in vaccinology agreed that the efficacy of the current COVID-19 vaccines at reducing the spread of SARS-CoV-2, disease complications and deaths far outweigh the likely risk of adverse events.

We have determined some heterogenous adverse events to COVID-19 vaccine’ administration in Africa. These heterogenous adverse events among Africans were statistically significant, and can be reported in three categories: uncommon, common, and very common signs as reported in the result section. These adverse events among Africans reported here align with similar ones reported from other continents including dizziness, fever, and tenderness at the injection site remain recurrent, while abdominal pain and menstrual disorder were less reported in other populations [18,20,21].

Nonetheless, transient local inflammation signaling neutrophils and antigen presenting cells to the site of injection is expected after intramuscular administration of lipid nanoparticle formulated mRNA vaccines. According to the CDC, following the administration of the first dose of the COVID-19 vaccine, if an itch, swollen or painful rash is observed, such person(s) should be treated with antihistamine or acetaminophen. If fatigue or pain is observed, treatment is equally recommended before such candidates proceed for the second shot of the vaccine based on availability to affirm complete protection [26]. In all cases, it becomes necessary to critically evaluate patients’ previous medical histories and vaccine-associated allergies in detail; it is also important to monitor vaccinated persons for at least 30 minutes following COVID-19 vaccine administration to ensure that no immediate untoward effects are observed.

Respondents from Nigeria, Ghana and Kenya contributed more to the survey (Table 6); whether this observation was due to willingness of the respondents to complete the questionnaire, the intensity of administration of the questionnaire, ability to access the questionnaires, or the effect of population distributions is unclear. However, the reasons for low participation in online surveys in Africa are obvious and multifaceted, including but not limited to the lack of power supply, poor access to or irregular internet and electronic gadgets, lack of motivation, perception of questionnaires as too burdensome or intrusive, unexplained context of the objectives, or ethical-related issues.

In this study, adverse events were managed and treated at home without any visit to the hospital (37%; n=361) or with the use of traditional remedy (1%; n=7). These observations agree with previous review [21] that only a limited proportion of patients who have serious adverse events elsewhere considered treatment. These may be a reflection of the following: 1) the under-resourced situation of most healthcare facilities in Africa; 2) the lack of healthcare service delivery where it is most-needed, particularly, among the poor, and 3) the costs associated with seeking healthcare in Africa, including the lack of health insurance for the majority of the populations. In addition, previous experiences of people seeking hospitalization including wrong or unfavorable evaluations and diagnoses by physicians, health care organisations, or the low perceived need to seek medical care with views that the illnesses or symptoms will improve over time, or time constraints to visit hospitals may also be major constraint [27].

The study participants were vaccinated for COVID-19, based on the various recommendations or knowledge channels (Figure 5), which are 5-10 times more enablers of COVID-19 vaccination compared to the least used sources (religious and community leaders (2.7%), social media (2.7%), not-for-profit organizations (NGOs) (0.8%), and celebrities (0.3%)); it is recommended that for the society to benefit optimally from COVID-19 vaccine coverage, earlier proposal of Anjorin et al. [15], on the use of multi-channel vaccination campaign strategy is necessary for excellent vaccine uptakes.

Round 2

Reviewer 3 Report

Thank you for your answers, I agree with them and it would be important that they were added to the manuscript